# Positive and Negative Expectations Associated with Coffee and Energy Drinks: Implications for Placebo–Nocebo Research in Sports

**DOI:** 10.3390/nu17040628

**Published:** 2025-02-10

**Authors:** Angéla Somogyi, Merve Uca, Celal Bulgay, Ernest Yeboah Acheampong, Ricardo de la Vega, Roberto Ruíz-Barquín, Attila Szabo

**Affiliations:** 1Department of Psychology and Health Management, Faculty of Health and Sport Sciences, Széchényi István University, 9026 Győr, Hungary; 2School of Physical Education and Sports, Istanbul Esenyurt University, 34510 Istanbul, Türkiye; merveuca@esenyurt.edu.tr; 3Sports Science Faculty, Bingol University, 12000 Bingol, Türkiye; cbulgay@bingol.edu.tr; 4Department of Health, Physical Education, Recreation and Sports, University of Education, Winneba P.O. Box 25, Ghana; 5Department of Physical Education, Sport and Human Movement, Autonomous University of Madrid, 28049 Madrid, Spain; ricardo.delavega@uam.es; 6Department of Developmental and Educational Psychology, Autonomous University of Madrid, 28049 Madrid, Spain; roberto.ruiz@uam.es

**Keywords:** athlete, caffeine, performance-enhancement, placebo, sports, training

## Abstract

Background: Nutritional expectations have been shown to influence exercise performance via placebo and nocebo effects. The present study aimed to evaluate performance-enhancement expectations for coffee and energy drinks using the Stanford Expectations of Treatment Scale. Methods: A total of 402 participants (48.5% male) with an average exercise history of 4.53 years, engaging in average physical activity 3.91 times per week, were included in the study. Data collection was conducted through the Qualtrics platform. Results: Participants exhibited significantly higher positive expectations for coffee compared to energy drinks (*p* = 0.002), whereas negative expectations were more pronounced for energy drinks than for coffee (*p* < 0.001). Males and individuals engaging in anaerobic exercise expressed more positive expectations for energy drinks than females and those participating in aerobic or mixed exercise regimens. Additionally, high-frequency exercisers (≥4 sessions per week) reported greater positive expectations for both beverages and fewer negative expectations for coffee than low-frequency exercisers (≤3 sessions per week). Correlational analyses revealed weak but significant associations between expectations and variables such as exercise history, frequency, intensity, and age. Conclusions: The findings suggest that coffee is perceived as a more effective performance enhancer and exhibits greater placebo-inducing potential than energy drinks, which may elicit stronger nocebo effects. These group-specific perceptions should be considered by trainers, coaches, and researchers when addressing placebo–nocebo mechanisms in the context of sports and exercise.

## 1. Introduction

Caffeine, a key component in coffee and many energy drinks, has been extensively documented for its performance-enhancing effects in sports and exercise. Meta-analytical evidence supports the ergogenic benefits of both coffee [1,2,3,4,5,6] and energy drinks [7,8,9,10]. Despite these findings, the World Anti-Doping Agency (WADA) does not classify caffeine as a prohibited substance in competitive sports. However, caffeine is listed in WADA’s monitoring program, reflecting ongoing scrutiny of its potential impact on athletic performance [3].

Caffeine has consistently been shown to exert placebo effects on sports and exercise performance [5,6,11,12,13]. These effects are driven by psychological mechanisms, particularly expectations formed through prior learning or exposure to credible information. Specifically, placebo effects arise from the interaction of anticipated outcomes and perceived efficacy [14], highlighting the critical role of expectations in the performance-enhancing properties of coffee and energy drinks [15,16]. Empirical studies further substantiate this phenomenon. For instance, a well-controlled study by Schneider et al. (2006) demonstrated that individuals who believed they had consumed caffeinated coffee reported heightened alertness, even when given decaffeinated beverages [17]. Similarly, another study revealed that participants who received positive information (induced expectations) about a fictitious green energy drink improved their 200 m sprint times by an average of 2.41 s and perceived their performance as superior compared to those exposed to neutral or negative information about the drink’s performance-enhancing properties [18].

These findings suggest that both coffee and energy drinks possess actual as well as expectancy-driven placebo effects. However, the relative strength of these expectancy effects between the two stimulants remains unclear, particularly in regular exercisers. Thus, identifying which of these substances elicits more robust expected performance-enhancing properties—and thereby functions as a more potent placebo agent—warrants further investigation. Notably, two systematic reviews on the placebo effect in exercise [19,20] indicate that coffee may generate more optimistic expectations, although data on its potential expected adverse effects remain sparse.

Given the established influence of expectancies on the relationship between stimulants and performance outcomes [21], investigating whether coffee and energy drinks elicit differing expectancy effects among exercisers holds both theoretical and practical significance. This research question addresses a critical gap in understanding how individuals perceive the potential positive and negative impacts of these stimulants on performance. Additionally, from an exploratory perspective, it is pertinent to examine whether these expectancy effects are associated with exercise-related characteristics, such as exercise history, frequency, intensity, and type. Furthermore, variations in expectancy effects based on demographic factors, including age and gender, warrant consideration. This preliminary cross-sectional study addresses these questions, providing a foundational understanding of the nuanced interplay between expectancies, stimulants, and exercise-related variables. The findings of this investigation may serve as a basis for future systematic research aimed at further elucidating the role of expectancy mechanisms in sports and exercise contexts.

## 2. Materials and Methods

### 2.1. Participants

Volunteer participants were recruited through English social media platforms, including LinkedIn, Facebook, Twitter, and Instagram, without geographical limitation. The snowball sampling method was also employed to enhance the sample size [22]. This method implies that participants share the study link with others possibly interested in participating. Participants’ mean age was 23.52 ± 8.08 years, ranging from 18 to 81. The inclusion criteria required participants to be 18 years or older with no upper age limit and to engage in regular weekly exercise. While the initial target was 500 participants, data collection was halted after obtaining 469 responses due to a lack of further responses despite repeated recruitment efforts (reposting of the call for participation).

Of the initial dataset, 54 responses were excluded due to missing age data, precluding eligibility verification. An additional 13 responses were removed because the survey was started but interrupted in less than three minutes, which yielded partial answers and insufficient time for valid responses. Therefore, the final sample comprised 402 participants, including 195 males (M_age_ = 23.60, SD = 7.28) and 207 females (M_age_ = 23.58, SD = 7.08).

Participants reported a mean exercise history of 4.53 years (SD = 1.69), a mean weekly exercise frequency of 3.91 sessions (SD = 1.785), and a mean self-reported exercise intensity of 4.57 (SD = 1.58) on a 1 to 10 Likert scale. Most respondents (n = 183) engaged in aerobic and anaerobic exercises (intervals of low, steady, and high-intensity training). At the same time, 144 participated exclusively in aerobic activities (maintaining steady-effort training), and 73 performed anaerobic exercises (rest or low-intensity exercise periods followed by very high-intensity short-duration exercises). These classifications were made with 100% agreement by three research team members.

The Research Ethics Board of Széchenyi István University, Győr, Hungary, granted ethical approval for the study (Permission No. SZE/ETr-14/2024 (X.3.)). All participants provided informed consent and completed the survey anonymously. The study adhered to the principles outlined in the Declaration of Helsinki for research involving human participants [23].

### 2.2. Materials

Demographic variables, including age, gender, and exercise habits, were assessed through a series of standardized questions. To evaluate positive and negative expectations regarding the effects of coffee and energy drinks on sports and exercise performance, the psychometrically validated Stanford Expectations of Treatment Scale (SETS) was employed [24]. We chose this instrument because no similar tools assess treatment-related expectations on a specific outcome. The SETS comprises six items, three measuring positive and three assessing negative expectations. Each item is rated on a seven-point Likert scale, ranging from strong disagreement to strong agreement.

In adapting the scale for this study, the term “treatment” was replaced with “coffee” and “energy drinks” to reflect the specific focus on these substances. Despite this modification, the internal consistency of the adapted scale remained high, with Cronbach’s alpha values exceeding 0.80 in all instances, consistent with the reliability reported in the original validation study [24]. This adaptation ensured that the scale was contextually relevant and psychometrically robust for the current research purposes.

### 2.3. Procedure

Participants completed the survey online via the Qualtrics research platform. After data collection, the dataset was exported into SPSS for analysis. The data underwent thorough verification and cleaning processes to ensure accuracy and completeness. The raw data are available at the Mendeley repository (DOI: 10.17632/yj5ncwhdh5.1).

Statistical analyses included normality assessments, bias-corrected bootstrapped Spearman correlations, and nonparametric tests to account for the ordinal nature of the data and potential deviations from normality. Specifically, Friedman’s Analysis of Variance (ANOVA) was used to evaluate within-group differences in expectations. Group differences were analyzed using the McNemar test for paired nominal data and the Kruskal–Wallis H exact test, accompanied by Monte Carlo simulations (10,000 samples; 99% confidence interval) to enhance robustness and precision. These advanced statistical techniques ensured rigorous examination of expectation differences across participant subgroups.

## 3. Results

Before the statistical analysis, we had to determine data normality. The Kolmogorov-Smirnov and Shapiro-Wilk tests indicated that the assumption of data normality was violated; consequently, nonparametric statistical methods were employed for analysis. First, bootstrapped Spearman’s rank correlations were conducted to assess the relationships between exercise characteristics (e.g., history, frequency, and intensity) and both positive and negative expectations regarding the effects of coffee and energy drinks on exercise performance. The results of these analyses are presented in Table 1.

Second, Bonferroni-corrected (to account for Type I error) nonparametric Friedman Analysis of Variance (ANOVA) tests were performed to examine differences between positive and negative expectations associated with coffee and energy drinks. Statistically significant differences were observed between positive and negative expectations for both coffee (positive expectation: M = 11.54, SD = 4.10; negative expectation: M = 9.19, SD = 4.28; Q = 5.62, *p* < 0.001) and energy drinks (positive expectation: M = 9.77, SD = 4.09; negative expectation: M = 12.96, SD = 4.53; Q = 7.04, *p* < 0.001). Additionally, positive expectations for coffee were significantly higher than for energy drinks (Q = 3.60, *p* = 0.002), while negative expectations for energy drinks were statistically significantly greater than for coffee (Q = 9.06, *p* < 0.001).

We re-tested a random subsample (n = 187) for statistical cross-validation purposes, and the results yielded the same findings. For example, using Bonferroni correction for multiple tests, statistically significant differences were observed in positive and negative expectations for both coffee (positive expectation: M = 11.57, SD = 4.06; negative expectation: M = 9.24, SD = 4.51; Q = 4.41, *p* < 0.001) and energy drinks (positive expectation: M = 9.57, SD = 4.13; negative expectation: M = 12.93, SD = 4.60; Q = 4.57, *p* < 0.001). Additionally, positive expectations for coffee were significantly higher than for energy drinks (Q = 2.69, *p* = 0.042), while negative expectations for energy drinks were statistically significantly greater than for coffee (Q = 6.28, *p* < 0.001).

To further analyze these findings, difference (delta [Δ]) scores between positive and negative expectations were calculated, with positive differences coded as 1 and negative differences as 2, reflecting the dominant expectation (positive or negative) for each drink. To generate the most reliable *p*-values, a bootstrapped McNemar test was applied to these categorical data, yielding a statistically significant chi-square value (χ^2^ = 10.46, *p* < 0.001). Among respondents, 65.6% exhibited a positive Δ score and 34.4% a negative Δ score for coffee-related expectations. In contrast, for energy drinks, 39.2% exhibited a positive Δ score, and 60.8% reported a negative Δ score (Figure 1).

Lastly, Kruskal–Wallis H exact tests with Monte Carlo simulations (10,000 samples, 99% confidence interval to improve *p*-value estimation) were used to evaluate group differences in expectations based on (1) gender, (2) type of exercise (aerobic, anaerobic, or mixed), and (3) weekly exercise frequency (high-frequency [≥4 (median) sessions/week] versus low-frequency [≤3 (below median) sessions/week]). We also employed the False Discovery Rate (FDR) method using the Benjamin–Hochberg (BH) procedure. Results indicated that individuals engaging in anaerobic exercise reported significantly greater positive expectations for energy drinks than those performing aerobic exercise. Similarly, males exhibited higher positive expectations for energy drinks than females. For coffee, high-frequency exercisers had significantly higher positive and lower negative expectations than low-frequency exercisers. These findings are summarized in Table 2, with descriptive statistics detailed in Table 3.

## 4. Discussion

The present study indicates that positive expectations regarding the ergogenic effects of coffee are significantly greater than those for energy drinks, whereas negative expectations are more pronounced for energy drinks. One possible explanation is that coffee has a long-standing cultural association with alertness and cognitive enhancement, while energy drinks may be viewed more as situational or recreational. Based on anecdotal information or subjective beliefs about their ingredients, some individuals might also have reservations about consuming energy drinks. In this context, participants may not be fully aware of the specific caffeine content in each beverage, which could influence their expectations. While our current questionnaire did not explicitly assess participants’ knowledge of caffeine content or their consumption patterns of either drink, we acknowledge this omission as a limitation that should be addressed in future research.

A weak inverse relationship was observed between exercise history and negative expectations associated with the performance-enhancing properties of both coffee and energy drinks. Conversely, habitual exercise intensity positively correlated with positive expectations for both beverages. A plausible explanation is that a longer exercise history, reflecting greater long-term commitment and involvement, may be associated with a stronger focus on maximizing exercise performance. This could lead to a more positive attitude toward nutrients or supplements that could enhance performance or reduce effort. While speculative, this explanation offers a potential avenue for future research.

In addition, weak but significant correlations were found between expectations, exercise frequency, and age. Specifically, exercise frequency showed a weak but significant positive correlation with positive expectations for both drinks and a negative correlation with negative expectations. Similarly, age had a weak but statistically significant positive correlation with positive expectations about coffee and a weak negative correlation with negative expectations for both drinks.

Males and individuals engaging primarily in anaerobic exercise exhibited more favorable expectations towards the ergogenic effects of energy drinks compared to females and those participating in aerobic or mixed forms of exercise. While speculative, it is possible that males and individuals engaging in anaerobic exercises, such as weightlifters and sprinters, tend to believe more in the effectiveness of energy drinks than other exercisers because these drinks, in addition to caffeine, also contain other stimulants that enhance short bursts of power, strength, and focus, which are critical for high-intensity, explosive movements.

High-frequency exercisers (≥4 sessions per week) reported greater positive expectations for the performance-enhancing effects of both coffee and energy drinks, alongside lower negative expectations for coffee. However, this trend was not observed for energy drinks. These findings underscore the importance of considering individual exercise characteristics, including history, frequency, intensity, and type, when examining the role of expectancy in exercise performance. Such factors may modulate the placebo and nocebo effects of ergogenic substances, warranting further investigation in future research.

The findings of this study suggest that while both coffee and energy drinks contain caffeine—and energy drinks may sometimes have a higher caffeine content—regular exercisers perceive coffee as having better performance-enhancing effects. An 8-ounce serving of coffee typically contains 80–100 mg of caffeine, whereas energy drinks vary from 40–250 mg per 8 ounces [25]. However, the ergogenic effects of caffeine are dose-dependent, with effective enhancement requiring approximately 6–9 mg/kg of body weight [25]. For an individual weighing 70 kg, this equates to 420–630 mg of caffeine, corresponding to roughly 10 servings of coffee or energy drinks. Given that such high levels of caffeine consumption are uncommon before exercise, participants’ expectations are likely shaped by global information sources, including social and media influences, rather than direct personal conditioning. Nevertheless, these effects could be shaped by conditioning effects as well. For example, regular and occasional coffee or energy drink consumers may have different expectations associated with these caffeine-containing drinks, which could affect these results. Regrettably, we did not assess the habitual consumption of these drinks. Therefore, future studies untangling conditioning effects from expectations (or controlling for them) are needed.

Expectations alone can significantly influence performance outcomes, as demonstrated by prior studies on coffee [17,26] and energy drinks [18]. In the current study, approximately two-thirds of participants expressed positive expectations regarding the ergogenic effects of coffee, compared to only 39.2% for energy drinks. Conversely, 60.8% of participants held negative expectations towards energy drinks, which are known to impair motor performance via nocebo effects [26]. These findings indicate that coffee is more likely to elicit placebo effects despite comparable caffeine content, while energy drinks are more prone to trigger nocebo effects. Moreover, expectation variations were modestly associated with exercise history, frequency, intensity, and age, emphasizing the importance of accounting for these factors in future investigations of placebo and nocebo effects in exercise contexts. Notably, many previous studies employing sub-therapeutic caffeine doses [25] may need to be reevaluated, as their findings could be attributable to expectancy effects rather than the physiological impact of coffee or energy drinks [27].

The results have important implications for placebo and nocebo research in exercise contexts. Although we did not observe placebo or nocebo effects in our study, the finding that coffee is associated with stronger positive expectations than energy drinks—and conversely, energy drinks are linked to more negative expectations—suggests that these beliefs are likely to influence exercise performance [22,28,29]. Coffee’s stronger association with positive expectations may position it as a more potent placebo agent, especially in contexts where psychological factors are particularly influential. In contrast, the negative expectations tied to energy drinks could lead to nocebo effects, where perceived adverse outcomes inhibit performance. This contrast warrants further examination, particularly concerning objective performance measures. Future research should investigate how these anticipated outcomes directly influence exercise performance, whether in aerobic tasks, such as time or distance, or anaerobic exercises, such as weight load or repetition count. Understanding these relationships is essential for fully exploring the role of expectations in performance enhancement and impairment.

The present study has some limitations. Its cross-sectional design precludes the establishment of causation. Additionally, the reliance on volunteer participants may have introduced selection bias, limiting the generalizability of the findings to the broader exercising population. Recruitment through social media may introduce selection bias (e.g., younger, higher socioeconomic status participants). Hence, future studies should use alternative recruitment methods for cross-validation and improved generalizability. Moreover, considering the impact of genetics on coffee metabolism, future research should take genetic factors into account. Since genetic variations can influence caffeine metabolism and individual responses, analyzing these molecular markers will contribute to obtaining more comprehensive and reliable results [30,31,32,33,34,35,36]. Furthermore, the study utilized generic formulations of coffee and energy drinks, but participants may have based their responses on preferences for specific brands, potentially influencing their expectations and perceptions. These limitations should be addressed in future research to enhance the robustness and applicability of the findings.

## 5. Conclusions

The present study underscores a significant divergence in the perceived effects of coffee and energy drinks on exercise performance, with coffee emerging as the more favorably regarded performance enhancer. The statistically significant differences in positive and negative expectations indicate that coffee is perceived to have more potent ergogenic effects among regular exercisers. Based on the expectancy model of placebo effects, coffee may also possess superior placebo-inducing properties compared to energy drinks, which should be explored via objective performance measures in future research. In contrast, with over 60% of participants reporting negative expectations about energy drinks, these perceptions may make individuals more susceptible to nocebo effects, especially in specific demographic groups, such as females and those participating in aerobic exercise.

These findings have practical implications for professionals in sports and exercise, including trainers, coaches, and nutritionists. Understanding the psychological influence of beverage choices on performance outcomes can inform recommendations. Promoting coffee as a preferred option could leverage its perceived ergogenic benefits while mitigating the adverse effects of negative expectations linked to energy drinks, potentially enhancing both exercise performance and overall well-being.

From a research perspective, the growing body of placebo and nocebo studies in sports and exercise science should account for the observed discrepancies in perceived efficacy between coffee and energy drinks. The selection of a placebo or nocebo agent in experimental designs should consider the perceived potency of these substances, as such perceptions may contribute to variations in the magnitude of observed effects. Indeed, inconsistencies in placebo and nocebo research outcomes [19] may partly be attributable to the differing expectancy profiles of the agents utilized [37]. Addressing these factors in future studies could provide greater clarity and enhance the reliability of findings in this field.

## Figures and Tables

**Figure 1 nutrients-17-00628-f001:**
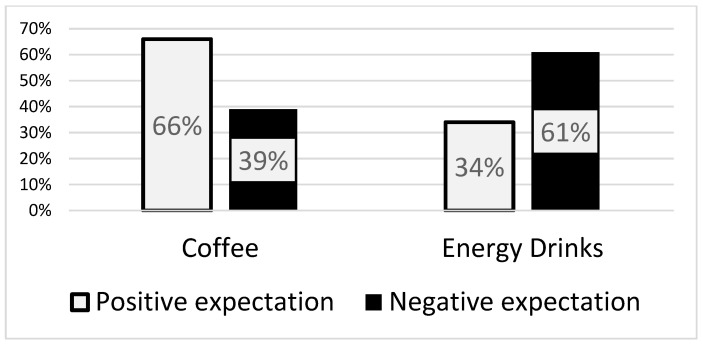
Dichotomized positive vs. negative expectations associated with coffee and energy drinks.

**Table 1 nutrients-17-00628-t001:** Bias-corrected bootstrapped Spearman’s correlations between exercise characteristics, age, and expectancies associated with performance-enhancing effects of coffee and energy drinks.

			PCF	NCF	PED	NED
Exercise history	Spearman’s Rho		0.122	−0.149 *	0.106	−0.227 *
	*p* (2-tailed)		0.055	0.019	0.095	<0.001
	Bootstrap bias		0.001	−0.001	0.002	0.000
	Std. Error		0.062	0.064	0.058	0.061
	BCa 95% CI	Lower	0.002	−0.267	−0.007	−0.342
		Upper	0.247	−0.033	0.227	−0.107
Exercise frequency	Spearman’s Rho		0.238 *	−0.225 *	0.184 *	−0.170 *
	*p* (2-tailed)		<0.001	<0.001	0.004	0.007
	Bootstrap bias		0.001	−0.002	0.001	0.001
	Std. Error		0.065	0.062	0.061	0.065
	BCa 95% CI	Lower	0.100	−0.349	0.067	−0.292
		Upper	0.364	−0.112	0.309	−0.043
Exercise intensity	Spearman’s Rho		0.187 *	−0.114	0.226 *	−0.11
	*p* (2-tailed)		0.003	0.073	<0.001	0.085
	Bootstrap bias		0.002	−0.003	0.000	0.000
	Std. Error		0.064	0.062	0.060	0.060
	BCa 95% CI	Lower	0.059	−0.234	0.104	−0.231
		Upper	0.316	−0.006	0.345	0.005
Age	Spearman’s Rho		0.228 *	−0.222 *	0.124	−0.145 *
	*p* (2-tailed)		<0.001	<0.001	0.052	0.023
	Bootstrap bias		−0.001	0.002	0.000	0.004
	Std. Error		0.059	0.061	0.065	0.065
	BCa 95% CI	Lower	0.116	−0.346	0.000	−0.268
		Upper	0.344	−0.095	0.252	−0.010

Note: *—statistically significant; BCa—Bias-Corrected accelerated; CI—Confidence Interval; PCF—positive expectations coffee; NCF—negative expectations coffee; PED—positive expectations energy drink; NED—negative expectation energy drink.

**Table 2 nutrients-17-00628-t002:** Group comparisons in positive and negative expectancies associated with coffee and energy drinks in exercise performance-enhancement using the Kruskal–Wallis H exact test with a Monte Carlo simulation (10,000 samples; 99% confidence interval).

Groups	K-W Test	PCF	NCF	PED	NED
Aerobic, anaerobic, and mixed (aerobic and anaerobic exercisers) **	H	0.850	1.275	10.711	2.223
	df #	2	2	2	2
	Monte Carlo’s exact *p*	0.649	0.531	0.005 *	0.331
	99% CI Lower	0.637	0.518	0.003	0.319
	99% CI Upper	0.661	0.543	0.006	0.343
High (≥4) and low (≤3) weekly exercise frequency groups	H	12.611	6.282	14.935	3.364
	df	1	1	1	1
	Monte Carlo’s exact *p*	<0.001 *	0.012 *	0.001 *	0.070
	99% CI Lower	0.000	0.009	0.000	0.063
	99% CI Upper	0.001	0.015	0.000	0.076
Males compared to females	H	1.797	0.298	8.206	3.295
	df	1	1	1	1
	Monte Carlo’s exact *p*	0.178	0.585	0.004 *	0.070
	99% CI Lower	0.168	0.573	0.002	0.063
	99% CI Upper	0.187	0.598	0.006	0.076

Note: # df—degrees of freedom; *—Statistically significant after adjusting the alpha (α) with the False Discover Rate (FDR) method using the Benjamin–Hochberg (BH) procedure; H—the value of the Kruskal–Wallis (K-W) statistic; CI—Confidence Interval; PCF—positive expectation coffee; NCF—negative expectation coffee; PED—positive expectation energy drink; NED—negative expectation energy drink; ** anaerobic exercisers differed from aerobic exercisers only (adjusted *p* = 0.003) in positive expectancy concerning the benefits of energy drinks on their exercise performance.

**Table 3 nutrients-17-00628-t003:** Means (totals of positive and negative expectations) and standard deviations in groups (Group column) and four types of expectations based on the Stanford Expectations of Treatment Scale (SETS).

	Group	n	Mean	SD
Positive Expectations for Coffee	aerobic	105	11.8286	4.16823
	anaerobic	38	11.1579	3.75986
	mixed	117	11.4274	4.14046
	Total	260	11.5500	4.09084
Negative Expectations for Coffee	aerobic	105	9.6286	4.57472
	anaerobic	38	8.9737	3.83784
	mixed	117	8.8205	4.10764
	Total	260	9.1692	4.26739
Positive Expectations for Energy Drink	aerobic	129	9.0775	4.05742
	anaerobic	61	11.2459	4.45592
	mixed	176	9.7557	3.86966
	Total	366	9.7650	4.09269
Negative Expectations for Energy Drink	aerobic	129	13.3101	4.93362
	anaerobic	61	12.3279	4.49341
	mixed	176	12.9148	4.23706
	Total	366	12.9563	4.53579
Positive Expectations for Coffee	Male	119	12.0336	4.15164
	Female	143	11.1259	4.02609
	Total	262	11.5382	4.10080
Negative Expectations for Coffee	Male	119	9.0084	4.20954
	Female	143	9.3427	4.34774
	Total	262	9.1908	4.28061
Positive Expectations for Energy Drink	Male	173	10.4393	4.20867
	Female	194	9.1649	3.88882
	Total	367	9.7657	4.08711
Negative Expectations for Energy Drink	Male	173	12.5260	4.43633
	Female	194	13.3454	4.58871
	Total	367	12.9591	4.52992
Positive Expectations for Coffee	Low exercise frequency	132	10.6136	4.15473
	High exercise frequency	130	12.4769	3.83823
	Total	262	11.5382	4.10080
Negative Expectations for Coffee	Low exercise frequency	132	9.8409	4.36822
	High exercise frequency	130	8.5308	4.10178
	Total	262	9.1908	4.28061
Positive Expectations for Energy Drink	Low exercise frequency	177	8.9322	3.76828
	High exercise frequency	190	10.5421	4.22713
	Total	367	9.7657	4.08711
Negative Expectations for Energy Drink	Low exercise frequency	177	13.4407	4.19119
	High exercise frequency	190	12.5105	4.79168
	Total	367	12.9591	4.52992

## Data Availability

The data presented in this study are openly available in a repository: doi:10.17632/yj5ncwhdh5.1.

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
