# Peer review of "Positive and Negative Expectations Associated with Coffee and Energy Drinks: Implications for Placebo–Nocebo Research in Sports"

_nutrients, 2025, doi:10.3390/nu17040628_

Round 1
Reviewer 1 Report
Comments and Suggestions for Authors
This survey-based study presents an interesting question regarding perceived benefits/detriments of energy drinks and coffee in regular exercisers. For the most part, the writing is clear and well-referenced. The literature is scant, however, in identifying potential cognitive and mental benefits regarding caffeine and the mechanism(s) potentially involved. The survey does not distinguish between individuals who are naïve vs. habitual users of caffeine? The rationale for the statistical approach used should be further detailed. Does the recruitment of subjects through social media imply a bias in selection of subjects (younger vs. older or socioeconomic status or educational level)? Is there a need for some cross-validation of this work? The Results presentation is the biggest limitation in the paper (besides providing improved statistical justification). The paper relies strictly on extremely dense Tables: Are there key variables that could be displayed with scatter plots or box plots across groups for more effective presentation (especially table 3)? This is particularly important when across group comparisons are being made (these are unclear in the presentation for Table 1 and table 3). Did the authors consider other validated survey tools beyond the SETS? This should be explained in either limitations or the rationale for methods selected.
Rationale for using a BOOTSTRAPPED McNemar should be made clear how it applies based on common definition: (unfamiliar if this is appropriate). The McNemar test is used to determine if there are differences on a dichotomous dependent variable between two related groups. It can be considered to be similar to the paired-samples t-test, but for a dichotomous rather than a continuous dependent variable.
Specific comments.
Line 87 Briefly Explain the snowball sampling method. There should be more details regarding whether this was limited geographically, and how repeated attempts were made. Also was there an age maximum “cut-off”?
Line 99 Define what was considered “anaerobic exercise”- was this strictly weight training or sprint interval work? This needs further clarification as resistance trainers may have different views than “endurance” slanted individuals.
Line 145. The following statement requires clarification: “Statistically significant differences were observed for both coffee (positive expectation: M = 11.54, SD = 4.10; negative expectation: M = 9.19, SD = 4.28; Q = 5.62, p < .001) and energy drinks.” Are you inferring that positive exceed negative values?
Line 163 I would not necessarily categorize 3 days a week (or less) as low (given they may have a longer duration and therefore exercise volume is not different)? Was a lower threshold cut point considered based on time spent per week instead of frequency?
Line 182 False Discover Rate (FDR) method using the Benjamin-Hochberg (BH) procedure appears in the Table caption but not listed in methods section? Please add to methods with a clear explanation/rationale.
Table 1. It is unclear how the sub-groups statistics are applied within each category. For example in Exercise History, is the p value a difference between the > 4 times per week vs. “low”. Table 2. For the grouping of type of exercisers there are 3 groups therefore the table fails to indicate which are different from each other (suggest create Figures with distribution of responses for each grouping) as alternative presentation.
Table 3- this needs to denote differences across the sub-groups. Please consider converting into Figures that show inter-group differences.
Line 192 where exactly is this result reported “ A weak inverse relationship was observed between exercise history and negative expectations associated with the performance-enhancing” properties of beverages
Line 214 This sentence is entirely speculative- is there any data available to back up this assumption?
Line 226 Age is not delineated to understand the group cut points analyzed?
Reviewer 2 Report
Comments and Suggestions for Authors
In their observational study, the authors examined the attitude of participants engaging in different sports activities toward coffee and energy drinks intake to enhance sports performance and evaluate the potential placebo and nocebo effects thereof. Moreover, they also aimed to see whether various training variables such as exercise type, duration, intensity, etc, as well as some demographic variables, impact the expected and perceived coffee and energy drinks effects on performance.
This is a nice short presentation although, in my opinion, it adds little to existing knowledge, as: 1. there are quite some papers addressing similar questions available, and 2. the conclusions are based on questionnaires which, as known are very subjective; it lacks any more objective parameters accounting for a more reliable interpretation and conclusion of the study, and due to some necessary missing data might be subjected to bias and consequently not representative.
The participants’ adherence should be stated in methods; i. e. what was the exact number of responders compared to all invited participants. In my opinion, this is an important fact that undoubtedly impacts the conclusions as it may derive bias and put into question the relevancy of the study and its conclusions which should not be generalized as the sample might not have been representative in case of low adherence; this, in my opinion, is a certain limitation of the study which is only scarcely mentioned.
Methods warrant additional description/clarification: The exact age of patients should be given (it is just stated older than 18); how did participants categorize the type of sport they were performing and how could authors be sure their categorization was reliable, based on what authors categorized participants regarding the sport type; also, the training intensity is not specified., etc.
Some additional data would be worth obtaining which at this stage is rather unlikely, namely: whether participants do consume coffee and/or energy drinks and if so, in which dose. This would be interesting information for further consideration and conclusions.
An inclusion of the questionnaire either in the text or as supplementary material in my opinion is mandatory.
An additional description of the delta score for differences between positive and negative expectations would be mandatory for clarity. A more thorough description of positive and negative expectations would be beneficial to the readers: which exactly they are and how they were scored: just 0 and 1 or also graduated. Also, the maximum score should be given; probably, column three in table 3 refers to this value? This should be stated. Moreover, potential speculation on why the correlation was more obvious for the participants involved in anaerobic sports would be nice in the discussion.
I’m not sure whether tab 3 is necessary as it gives no added value; it is not quite clear what column three (means of which value: total questionnaire score?) refers to. Additionally, if data were not normally distributed (as stated by the authors), they should be presented as median and not mean, etc. in table 3; there is partly duplication of data from table 2. The abbreviation clarification df used in tables is missing (probably difference?)
In the discussion, the authors mention that although both coffee and energy drinks contain caffeine as a potential ergogenic substance, the ergogenic and enhancing effects are rather expected for coffee among participants. A few words on speculation about why this is so would be nice; probably, the participants are not quite familiar with the content of coffee and drinks; respectively, this question should also be addressed and included in the questionnaire.
In the discussion, some additional speculation on the facts impacting the results and participants’ attitudes toward taking one or another should be given, and some psycho-social background of authors’ speculation/observation at least indicated. As already stated above, authors should speculate on potential psychological and/or physiological reasons why the correlations were greater in participants involved in anaerobic sports.
Last but not least, the study would gain much more relevance if some objective indices were gathered and correlated with expectations and realistically measured variables contributing to and accounting for sports performance.
Reviewer 3 Report
Comments and Suggestions for Authors
The authors report a study results on evaluating the impact of performance-enhancement expectations for coffee and energy drinks on exercise performance, using the Stanford Expectations of Treatment Scale. Results showed that participants had more positive expectations for coffee and more negative expectations for energy drinks, with differences based on gender, exercise type, and frequency. The findings suggest that coffee is perceived as a stronger performance enhancer with greater placebo potential, while energy drinks may induce stronger nocebo effects, highlighting the importance of considering these perceptions in sports and exercise contexts.
The study was well organized and results are analysed statistically. Of course, in such a study any result is a positive result but the reported findings may have practical implications for sports and exercise professionals, such as trainers, coaches, and nutritionists.
My very minor suggestion is to include word „average“ in „engaging in physical activity 3.91 times per week“ (line 21).
The manuscript is written in a good quality English, thus the only place I can suggest revising is Lines 251-252: the sentence should be revised to be clearly understandable.
